# A lithium-ion batteries SOH estimation method based on extracting new features during the constant voltage charging stage and improving BPNN

Yanhua Xian[1,2], Mingyang Li[1,2]*, Jiayin Huang[1,2]

**1** Guangxi Key Laboratory of Brain-inspired Computing and Intelligent Chips, School of Electronic and Information Engineering, Guangxi Normal University, Guilin, China, **2** Key Laboratory of Nonlinear Circuits and Optical Communications (Guangxi Normal University), Education Department of Guangxi Zhuang Autonomous Region, Guilin, China

* 2498406037@qq.com

## Abstract

Existing state of health (SOH) estimation methods for lithium-ion batteries predominantly extract health features (HF) from constant current (CC) and constant voltage (CV) charging phases. Nevertheless, CC charging phase feature extraction is susceptible to the randomness of the initial charging stage. By contrast, data during the constant voltage (CV) charging stage are preserved intact. The complexity and noise interference of battery data make it difficult to accurately extract health features, and it is necessary to develop effective methods to process the data and extract representative features. In response to this issue, this paper proposes an SOH estimation method for extracting HF at the end of the CV charging stage and optimizes the Backpropagation Neural Network (BPNN). Firstly, the current curve during the CV charging stage was transformed into the differential current curve (dQ/dI curve), from which two HFs were extracted. Secondly, addressing the issue of weight and threshold initialization in BPNN, the Coati Optimization Algorithm (COA) was employed to optimize the network (COA-BPNN). Finally, validation was conducted using two publicly available datasets. The experimental results demonstrate that the proposed method exhibits high accuracy in estimating the SOH of batteries under various environmental temperatures and charging rate conditions. Compared with the traditional BPNN method, the COA-BPNN method reduces the maximum root mean square error and average absolute error of the estimated results to 0.22% and 0.16%, respectively.

**Data availability statement:** All relevant data are within the paper and its Supporting Information files.

**Funding:** National Natural Science Foundation, China (6197022931). The funders had no role in study design, data collection and analysis, decision to publish, or preparation of the manuscript.

**Competing interests:** The authors have declared that no competing interests exist.

## Introduction

Lithium-ion batteries require a Battery Management System (BMS) with functions such as fault diagnosis, power management control, safety, and health status estimation to fully utilize their performance characteristics [1,2]. SOH is a critical parameter for lifecycle faults and safety warnings of lithium-ion batteries, reflecting the degree of battery aging and performance degradation [3]. The more precise the estimation of battery SOH, the more assured the safe and reliable operation of lithium-ion batteries [4].

In recent years, two main approaches to predicting battery SOH have emerged: model-based and data-driven methods. Significant progress has been made in improving the accuracy and stability of predictions using these approaches. Model-based approaches primarily represent battery physical and chemical structures using mathematical equations [5,6]. Tran et al. [7] proposed a comprehensive equivalent circuit model (ECM) that considers the influence of battery SOH, state of charge, and temperature on model parameters. Once the ECM's parameters are determined, they can be combined with filtering algorithms to achieve accurate SOH estimation. Ning et al. [8] constructed an adaptive battery model that accurately reflects the actual characteristics of batteries under various operating conditions. Model-based approaches have some limitations, including the complexity of model selection and the high dependency on the accuracy of model parameters.

In contrast, data-driven methods can evaluate battery SOH based on available external input parameters without relying on precise mechanistic models. Therefore, previous studies have widely adopted a variety of typical machine learning algorithms, including support vector machine (SVM) [9], relevance vector regression [10], and Gaussian process regression [11]. Additionally, various neural network models have been widely applied to battery SOH estimation, such as BPNN [12], long short-term memory neural networks [13] and convolutional neural networks [14] have also been utilized for battery SOH estimation. BPNN is a nonlinear model-based method that can exploit a system's underlying correlations through small-sample training when information about the nonlinear system is unknown. However, it also encounters issues regarding parameter initialization [15]. Guo et al. [16] proposed a method for obtaining the weights and thresholds of BPNN using the Sparrow Search Algorithm (SSA). Zhang et al. [17] optimized the weights and thresholds of BPNN by combining the Firefly Algorithm and applied the model's predicted outputs to the K-means algorithm for clustering. However, integrating multiple algorithms may lead to relatively complex models, which could increase the demand for computational resources. In the recent rise of Physical Information Neural Network (PINN) research, scholars have devoted themselves to combining physical laws with neural networks, providing new ideas and methods for battery health state estimation. PINN incorporates physical equations as soft constraints into the loss function of neural networks, allowing the network to not only learn statistical laws of data during training, but also follow physical laws, thereby improving the accuracy and generalization ability of the model. Relevant scholars have achieved significant research results in this field [18].

To ensure the accuracy of battery SOH estimation, selecting appropriate HFs and employing rational machine learning or deep learning algorithms is equally important. In recent years, numerous researchers have focused on extracting HFs at different stages of battery charging, such as constant-current (CC) charging, constant-voltage (CV) charging, and constant-current discharging. Due to road conditions and car driving's power demand-side response, battery discharge process data is uncertain [13], so HFs are usually extracted from the charging process. Features extractable during the CC charging stage include CC charging duration [19,20], geometric characteristics of the voltage curve [21,22], and charging capacity within voltage intervals [23]. Features extractable during the CV charging stage include CV charging duration [24], charging capacity [25], as well as charging current and differential current [26,27].

Although the estimation above methods based on HFs can yield relatively ideal battery SOH estimation results, they are hardly applicable in practical situations due to various influencing factors, such as the requirement for complete discharge or charge of the battery, which is almost impractical. Most users tend to charge the battery before it is fully depleted, resulting in a strong randomness in the initial charging stage [28]. Therefore, HFs extracted during the CC charging stage may lead to inaccurate estimation of battery SOH. In contrast, charging data during the CV charging stage are fully preserved and remain unaffected by the randomness of the initial charging stage. Wang et al. [29] proposed a novel HF based on the CV charging stage to estimate battery SOH under incomplete charging cycle conditions, but it requires the construction of complex equivalent circuit models. Ko et al. [30] introduced the differential current curve (dQ/dI curve) during the CV charging stage and utilized it to reflect the battery's health status. However, the investigation into battery SOH estimation in this study remains limited. Due to the varied charging habits among users, it is often difficult to precisely determine the initial stage of the charging process. However, most users tend to charge the battery to a State of Charge (SOC) of 100% [29]. With the rapid development of the electric vehicle industry, significant progress has been made in fast charging technology for lithium-ion batteries. This technology reduces the time required to charge the battery to full capacity and thereby enhances the user charging experience [31,32].

To address the issues above, this study proposes a lithium-ion batteries SOH extraction method by extracting HF at the end of the CV charging stage and improving BPNN. It aims to ensure the stability of the HF extraction process and the accuracy of the estimation results. The main contributions of this paper are as follows:

(1) This study pioneers a novel method for lithium-ion battery health feature extraction within the 200mA threshold during the terminal phase of CV charging;

(2) The initialization of weights and thresholds of BPNN is optimized using COA to enhance the stability of the network;

(3) Two publicly available datasets are used for validation, and the universality of SOH estimation is discussed. Experimental results demonstrate that the proposed method performs well on batteries at different temperatures and charging rates.

## Data analysis and feature extraction

### Battery datasets

This study validates the effectiveness of the proposed methods using the commercial 18650 battery datasets NCM and NCA provided in reference [33]. The battery is represented as CYX-Y/Z#N, where X represents temperature, Y represents charge C-rate, Z represents discharge C-rate, and N is the battery label. The study focuses on utilizing the following battery label from the datasets: CY35-0.5/1#1, CY35-0.5/1#2, CY45-0.5/1#9, CY45-0.5/1#10 from the NCM dataset, and CY25-0.25/1#1, CY25-0.25/1#2 from the NCA dataset. The batteries in the NCM dataset are charged at a constant current rate of 0.5C until the cut-off voltage of 4.2V is reached. Subsequently, they are maintained at a constant voltage of 4.2V until the current reaches 0.05C. The distinction lies in the ambient temperatures: batteries #1 and #2 operate at 35°C, while batteries #9 and #10 operate at 45°C. The NCA dataset batteries are charged at a constant current rate of

0.25C, with an ambient temperature of 25°C. The current sampling rate during the constant voltage charging stage for these batteries is 10s.

**dQ/dI curve**

Fig 1a illustrates the current variation curve of lithium-ion batteries during the constant voltage charging stage at different cycle counts. It is observed that with an increase in cycle counts, the required time gradually prolongs, indicating a significant correlation between the current during the constant voltage charging stage and the battery's SOH. This study employs analysis using the dQ/dI curve to more accurately extract the battery's aging characteristics. According to Eq. 1, I/(dI/dt) (or dQ/dI) represents the ratio of the charging current to its decay rate. Ref. [30] assigned a clear physical meaning to this curve through an equivalent circuit model, further establishing its effectiveness and rationality in battery aging analysis.

$$\frac{dQ}{dI} = \frac{dQ/dt}{dI/dt} = \frac{I}{dI/dt}$$

(1)

Transforming the original current curve of the constant voltage charging stage into the dQ/dI curve mainly involves the following three steps:

Step 1: According to the current integration Eq. 2, convert the current-time relationship in the original constant voltage charging stage into a capacity-time relationship;

$$Q(t) = \int_0^t I(\tau)d\tau$$

(2)

Step 2: By taking the current values during the constant voltage charging phase as the new independent variable, the original capacity-time relationship can be transformed into a capacity-current relationship, thus transforming the independent variable from time to current.

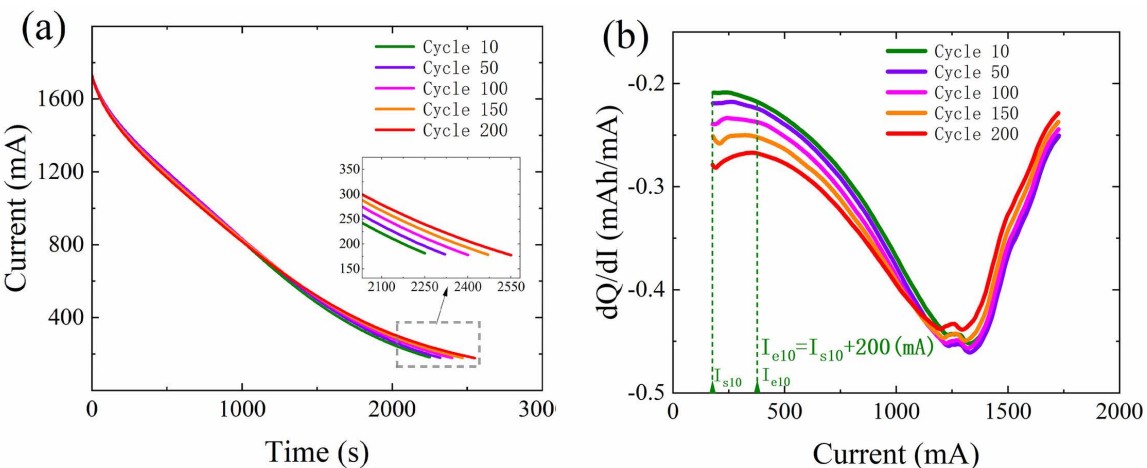

**Fig 1. Analysis of raw current data and dQ/dI curve.** (a) Current variation curves of battery CY45-0.5/1#9 in different cycles (b) dQ/dI curve of battery CY45-0.5/1#9 in different cycles.

Step 3: In the capacity-current relationship, by taking the first derivative of the capacity values, the dQ/dI curve can be obtained, as shown in Fig 1b. Some noise issues may arise during the generation process of the dQ/dI curve. This study employs the wavelet transform method for processing to reduce noise and smooth the curve, thereby improving the data's signal-to-noise ratio and the analysis's accuracy [34].

## Feature extraction

Variations in external battery parameters, such as voltage, current, and temperature, can reflect the change in SOH of lithium-ion batteries. These parameter changes provide the theoretical basis for extracting features from battery charge-discharge data and estimating SOH. The CC and CV charging curves contain a wealth of information regarding the battery's health status [27]. Due to variations in users' charging habits, it is often difficult to determine the initial stage of the charging process in daily life. This may lead to inaccuracies in extracting battery HFs during the CC charging stage. The HFs extracted from the CV charging stage can effectively mitigate the influence of randomness in the initial charging phase. This is because, during the constant voltage charging stage, the charging data of the battery are fully preserved, unaffected by the randomness at the beginning of charging. Therefore, extracting HFs during the CV stage allows for a more accurate estimation of the battery's SOH. As shown in Fig 1a, with the increase in battery cycle count, the current curve at the end of the CV charging stage exhibits a specific regular shift. Although users have varying charging habits, typically charge to a 100% SOC [29]. Therefore, this study selected a portion of the data from the end of the CV charging stage to estimate the battery SOH, making the HF extraction process more stable. As shown in Fig 1b, to more accurately capture the aging information of the battery, this study converted the current curve of the CV charging stage into a dQ/dI curve and extracted two HFs from it:

(1) The sum of dQ/dI data points ($SQI$) at the end of the constant voltage charging stage:

$$SQI_i = \sum_{j=1}^{N} (dQ/dI)_j$$

(3)

where $i$ represents the cycle period, $N$ is the total number of data points from the current value $I_{si}$ at the end of the CV charging stage to the current value $I_{ei}$ at 200mA before the end of the CV charging stage. Due to the discrete nature of the data, $I_{ei}$ can only be approximated.

(2) Integration of dQ/dI data points with respect to current ($\Delta Q$) at the end of the constant voltage charging stage:

$$\Delta Q_i = \int_{I_{si}}^{I_{ei}} (dQ/dI)dI$$

(4)

## Correlation analysis

Fig 2 illustrates the relationship between HFs and capacity with cycle periods. Both HFs exhibit significant linear correlations with cycle periods: $SQI$ decreases with increasing cycle numbers, while $\Delta Q$ increases.

This study employed Pearson correlation coefficient analysis [21] to validate the effectiveness of the extracted HFs. The correlation coefficient ranges from [−1, 1], with values closer to the absolute value of 1, indicating a stronger linear correlation between the HFs and battery capacity.

Table 1 presents the specific values of the correlation coefficients. The data in the table reveal a strong correlation between the extracted HFs and the actual SOH values, with the absolute values of the correlation coefficients all exceeding 0.94. Therefore, the extracted HFs can be used as input parameters for constructing predictive models.

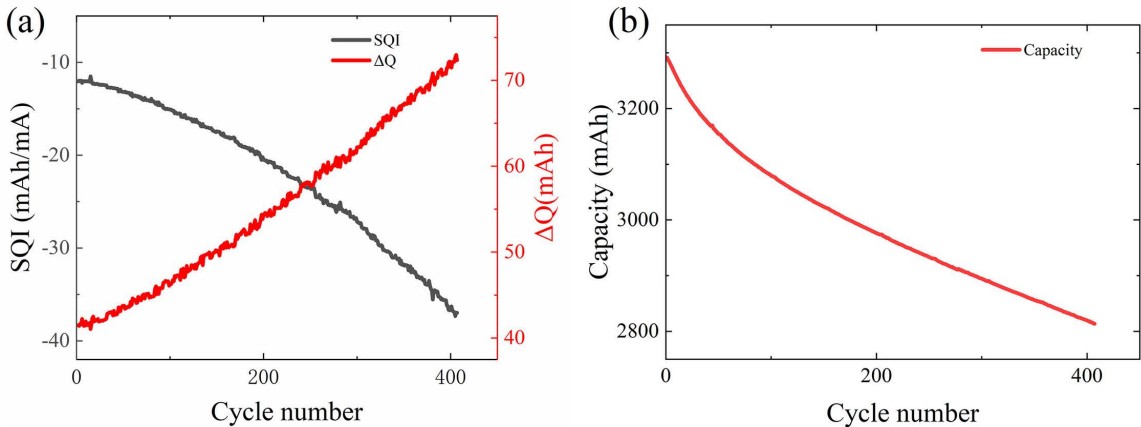

**Fig 2. HFs and capacity.** (a) The relationship between the HFs of battery CY45-0.5/1#9 and the cycle periods (b) the relationship between the capacity of battery CY45-0.5/1#9 and the cycle periods.

**Table 1. Pearson correlation coefficient of the extracted health features.**

| Datasets | Battery label | Pearson correlation coefficient | |
|---|---|---|---|
| | | *SQI* | *ΔQ* |
| NCM | #1 | 0.9547 | −0.9746 |
| | #2 | 0.9578 | −0.9820 |
| | #9 | 0.9433 | −0.9626 |
| | #10 | 0.9405 | −0.9596 |
| NCA | #1 | 0.9687 | −0.9835 |
| | #2 | 0.9631 | −0.9649 |

## Improved BPNN model

BPNN has strong generalization capabilities and robustness, and it typically performs well when dealing with noise and outliers in the data. However, when using BPNN, the initial parameters of the network are set randomly, which may cause the gradient descent algorithm to become trapped in local minima rather than global minima, thereby affecting the network's performance [15]. To address this issue, this study employs COA to initialize the weight and threshold parameters of BPNN, thereby improving the accuracy and robustness of the prediction results. The overall framework diagram is depicted in Fig 3.

## Principle of BPNN algorithm

BPNN is a shallow feedforward neural network model based on supervised learning. It minimizes the error between network predictions and actual values by adjusting the weights and thresholds within the network. BPNN employs gradient descent as its learning rule. A 3-layer BPNN can meet design accuracy requirements and achieve approximation of arbitrary nonlinear functions [17]. Therefore, this study chooses a 3-layer BPNN for predicting the SOH of lithium-ion batteries. In BPNN, the forward propagation of information first occurs from the input layer to the hidden layer, and then continues from the hidden layer to the output layer. As shown in Fig 3, the output of the hidden layer is denoted as $Y_j$, the threshold of the hidden layer is denoted as $\theta_j$, the threshold of the output layer is denoted as $\theta_k$, the weights from the input layer to the hidden layer are denoted as $\omega_{ij}$, and the weights from the hidden layer to the output layer are denoted as $\omega_{jk}$.

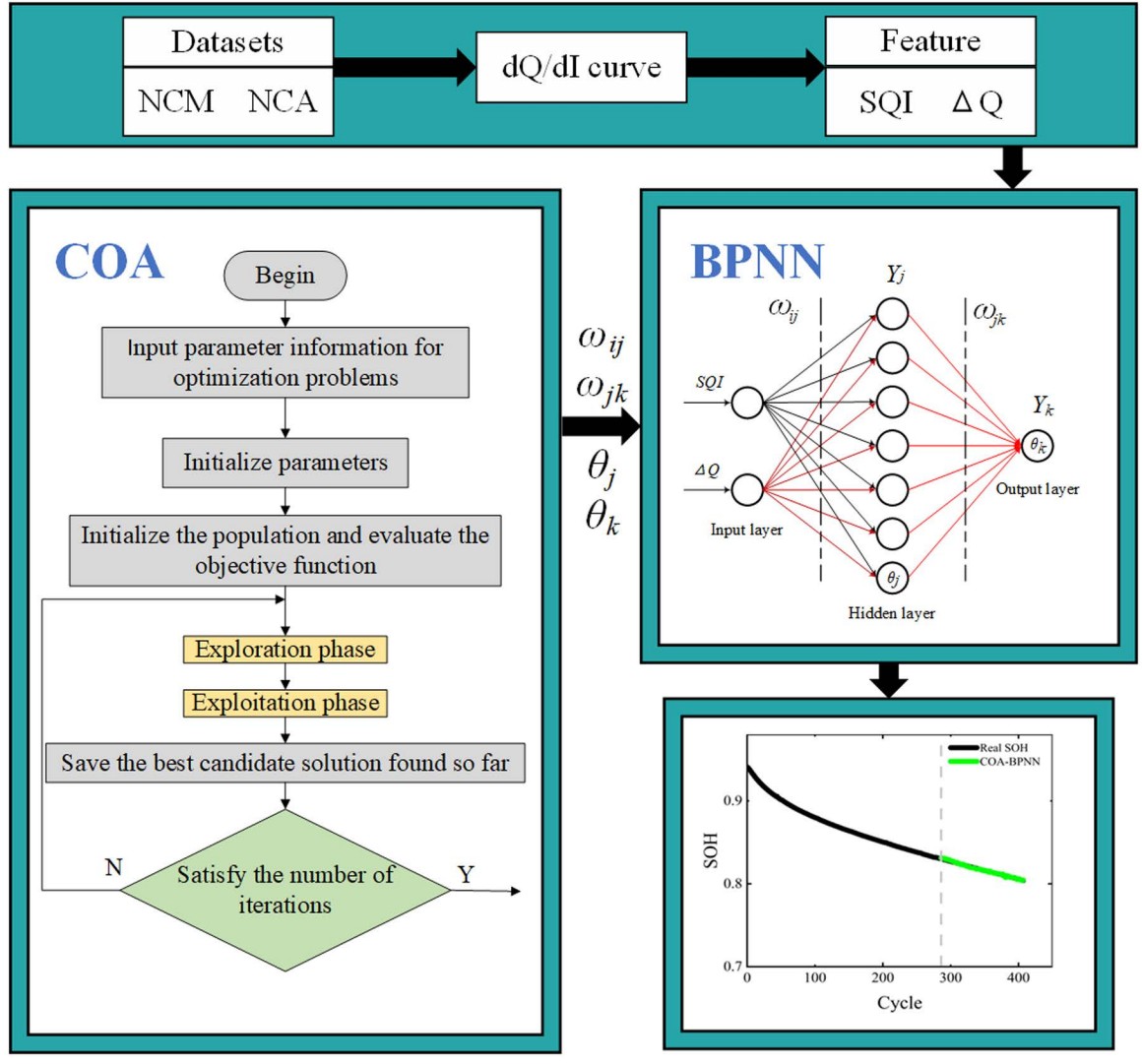

**Fig 3. Overall framework diagram.**

The error function in BPNN is used to measure the discrepancy between the network's predicted values and the actual values, denoted by $e$:

$$e = \frac{1}{N} \sum_{k=1}^{N} (T_k - Y_k)^2$$

(5)

where $N$ is the number of samples, $T_k$ is the expected output, and $Y_k$ is the network's output.

## COA principle

The COA method is a population-based metaheuristic algorithm, where coatis are regarded as population members [35]. The basic inspiration for the COA method comes from the strategies coatis employ when attacking iguanas and their

behavior simulations when facing and evading predators in nature. The algorithm demonstrates significant advantages over most intelligent optimization algorithms in balancing global exploration and local exploitation, rendering it more competitive. For example, SSA may require more iterations to converge, especially when optimizing high-dimensional complex problems, which may affect its convergence speed [16]. Due to its unique search strategy and adaptive parameter adjustment mechanism, COA usually has a faster convergence speed in the optimization process. It can find high-quality solutions in a short number of iterations, which is particularly important for BPNN parameter optimization problems that require a large amount of computation and iteration. In the COA algorithm, the position of each coati in the search space corresponds to a set of values for the decision variables. Therefore, the position of coatis represents a candidate solution to the problem. The COA principle mainly consists of three parts: (1) population initialization; (2) Hunting and attacking strategy on iguana (exploration phase); (3) The process of escaping from predators (exploitation phase).

## COA-BPNN

As shown in Fig 3, inputting the COA-optimized weights and threshold parameters into BPNN can mitigate the influence of the randomness of the network's initial parameters. Given that the BPNN has 3 layers, with 2 neurons in the input layer, 7 neurons in the hidden layer, and 1 neuron in the output layer. Hence, the total number of parameters in the BPNN network, including $\omega_{ij}$, $\omega_{jk}$, $\theta_j$, and $\theta_k$, is 29. The specific optimization steps are as follows:

Step 1: Input the parameters of the optimization problem, including the population size, number of iterations, total parameters of the BPNN network, upper and lower bounds for the network weights and thresholds, and the objective function to be optimized.

Step 2: Set the population size $N$ and the maximum number of iterations $T$.

Step 3: Initialize the population and evaluate the objective function, which involves computing the fitness values of coatis. Through $T$ iterations, identify the current best candidate solution. Here, the objective function is determined by the error value of the BPNN estimation results, as dictated by Eq. 5.

Step 4: Utilize $N/2$ of the population for the exploration phase. Calculate the new position of the $i$th coati according to Eq. 6, ensuring that the new position remains within the predefined bounds.

$$X_i^{P1} : x_{i,j}^{P1} = x_{i,j} + r \cdot (\mu_j - I \cdot x_{i,j}), i = 1, 2, \cdots, \frac{N}{2}; \quad j = 1, 2, \cdots, m \tag{6}$$

where $X_i^{P1}$ represents the new position of the $i$th coati, $x_{i,j}^{P1}$ represents its value in the $j$th dimension, $\mu_j$ represents the value of iguana in the $j$th dimension of the search space, $I$ is a randomly selected integer from the set {1,2}, $r$ is a random real number selected from the interval [0,1], $x_{i,j}$ represents the value of the $j$th decision variable, $N$ is the population size, $m$ is the number of decision variables.

Then, according to Eq. 7, compute the fitness value of the new position, and update the position of the $i$th coati based on the fitness value.

$$X_i = \begin{cases} X_i^{P1}, F_i^{P1} < F_i, \\ X_i, \ else \end{cases} i = 1, 2, \cdots, N \tag{7}$$

where $X_i$ is the position of the $i$th coati in the search space, $F_i^{P1}$ is the value of its objective function.

Then, continue exploration with the other half of the population. Generate a random position for the iguana according to Eq. 8. Next, compute the new position of the $i$th coati according to Eq. 9, and update the position of the $i$th coati based on Eq. 7.

$$\mu^G : \mu_j^G = lb_j + r \cdot (ub_j - lb_j), \ j = 1, 2, \cdots, \ m \tag{8}$$

$$X_i^{P1} : x_{i,j}^{P1} = \begin{cases} x_{i,j}+r\cdot(\mu_j^G-I\cdot x_{i,j}), & F_{\mu^G}<F_i, \\ x_{i,j}+r\cdot(x_{i,j}-\mu_j^G), & else \end{cases}$$
$$i = \frac{N}{2}+1, \frac{N}{2}+2, \cdots N; \ j = 1,2,\cdots, m \tag{9}$$

where $\mu^G$ represents a randomly generated position for the iguana after landing, $\mu_j^G$ represents its value in the $j$th dimension, $F_{\mu^G}$ represents the value of its objective function, $ub_j$ and $lb_j$ respectively represent the upper bound and lower bound of the $j$th decision variable.

Step 5: Utilize the population $N$ for the exploitation phase, simulating the behavior of coatis escaping predators. Calculate the upper and lower bounds for local search and the new position of the $i$th coati according to Eqs (10) and (11).

$$lb_j^{\varphi} = \frac{lb_j}{t}, ub_j^{\varphi} = \frac{ub_j}{t}, t = 1,2,\cdots, T \tag{10}$$

$$X_i^{P2} : x_{i,j}^{P2} = x_{i,j} + (1-2r)\cdot(lb_j^{\varphi} + r\cdot(ub_j^{\varphi} - lb_j^{\varphi})),$$
$$i = 1,2,\cdots, N; \ j = 1,2,\cdots, m \tag{11}$$

where $lb_j^{\varphi}$ and $ub_j^{\varphi}$ respectively represent the local lower bound and local upper bound of the $j$th decision variable, $X_i^{P2}$ represents the new position of the $i$th coati in the second phase, $x_{i,j}^{P2}$ represents its value in the $j$th dimension.

Then, compute the fitness value of the new position according to Eq. 12, and update the position of the $i$th coati based on the fitness value.

$$X_i = \begin{cases} X_i^{P2}, & F_i^{P2}<F_i, \\ X_i, & else \end{cases}, \ i = 1,2,\cdots, N \tag{12}$$

where $F_i^{P2}$ is the value of its objective function.

Step 6: Save the current best candidate solution found. If the iteration count has not reached $T$, return to Step 4 until the predetermined number of iterations is reached.

Step 7: Transmit the globally optimal position found by COA to the BPNN. Set the optimal parameters as the initial weights and thresholds of the BPNN, thus avoiding the uncertainty of initializing random parameters in the BPNN. Finally, the BPNN completes simulation training based on the set parameters and input health characteristics ($SQI$ and $\Delta Q$), and outputs the predicted values.

## Experiments and analysis

In this section, the extracted HFs will be experimentally validated using batteries subjected to different temperatures and charging rates. Additionally, the proposed COA-BPNN will be compared with traditional BPNN and other models in estimating the SOH of lithium-ion batteries, to validate the generalization of the proposed method.

## Evaluation criteria

In this study, the SOH is defined as the ratio of the current maximum capacity to the rated capacity, as shown in Eq. 13:

$$SOH = Q_c/Q_r \tag{13}$$

where $Q_c$ is the maximum capacity within the current cycle, and $Q_r$ is the rated capacity of the battery.

To evaluate the performance of the prediction models, this study uses the mean absolute error (MAE), root mean square error (RMSE), and the coefficient of determination $R^2$ to assess the estimation results of SOH. Where a result closer to 1 for $R^2$ indicates a better fit between the predicted values and the actual values.

$$MAE = \frac{1}{N}\sum_{k=1}^{N}|(Y_k - T_k)| \tag{14}$$

$$RMSE = \sqrt{\frac{1}{N}\sum_{k=1}^{N}(Y_k - T_k)^2} \tag{15}$$

$$R^2 = 1 - \frac{\sum_{k=1}^{N}(Y_k - T_k)^2}{\sum_{k=1}^{N}(T_k - \overline{T}_k)^2} \tag{16}$$

where $\overline{T}_k$ represent the mean of expected output, and $k$ represents the cycle number.

## Estimation results and analysis of SOH

In this study, a subset of batteries from datasets NCM and NCA is used to validate the effectiveness of the proposed method. The first 70% of the data is selected as the training set, while the remaining 30% is used as the testing set. The two HFs extracted during the CV charging phase are used as inputs for the model, while the battery's SOH serves as the output.

### Prediction of battery SOH at different temperatures

To validate the applicability of the proposed method for estimating the SOH of batteries at different environmental temperatures, this study utilizes CY35-0.5/1#1 and CY35-0.5/1#2 from the NCM dataset with an ambient temperature of 35°C, and CY45-0.5/1#9 and CY45-0.5/1#10 with an ambient temperature of 45°C as test subjects. The estimation results and errors of these batteries under 45°C and 35°C environments are shown in Fig 4.

According to Fig 4, it can be observed that the errors estimated by COA-BPNN are smaller than those of the other three algorithms. It is worth noting that COA-BPNN exhibits significant errors during the 1005th and 1012th cycle periods when estimating Battery CY35-0.5/1#2. This phenomenon is due to fluctuations in the recorded current data during the CV charging phase for these two cycle periods in the original dataset. Such anomalies may indicate potential safety hazards in the battery or faults in the BMS system.

The comparison results of these batteries in different models are shown in Table 2. According to Table 2, it can be observed that SVM, BPNN, SSA-BPNN, and COA-BPNN achieve good estimation results for batteries at different environmental temperatures. The RMSE for four models is less than 0.37%, the MAE is less than 0.24%, and the $R^2$ value is greater than 0.9. However, the estimation performance of SVM is poor, with the error gradually increasing as the number of cycles increases. Its maximum RMSE is 0.37%, and its maximum MAE is 0.24%. The main reason is that SVM's performance is largely dependent on the choice of kernel function and the setting of parameters. Compared to traditional BPNN and SSA-BPNN, COA-BPNN achieves the best estimation results, with RMSE consistently less than 0.16%, MAE consistently less than 0.10%, and R2 consistently above 0.98. The results validate the significant effectiveness of COA in optimizing the initial parameters of BPNN, while also demonstrating the superiority of COA over SSA. Therefore, the proposed model not only maintains high accuracy in SOH estimation but also identifies anomalies in lithium-ion batteries or BMS systems, demonstrating the robustness of the proposed model across batteries at different environmental temperatures.

### Prediction of battery SOH at different charging rates

Since all batteries in the NCM dataset are charged at a rate of 0.5C, CY25-0.25/1#1 and CY25-0.25/1#2 from the NCA dataset, charged at a rate of 0.25C and with an ambient temperature of 25°C, are used as experimental subjects. The estimated results and errors of these two batteries are shown in Fig 5.

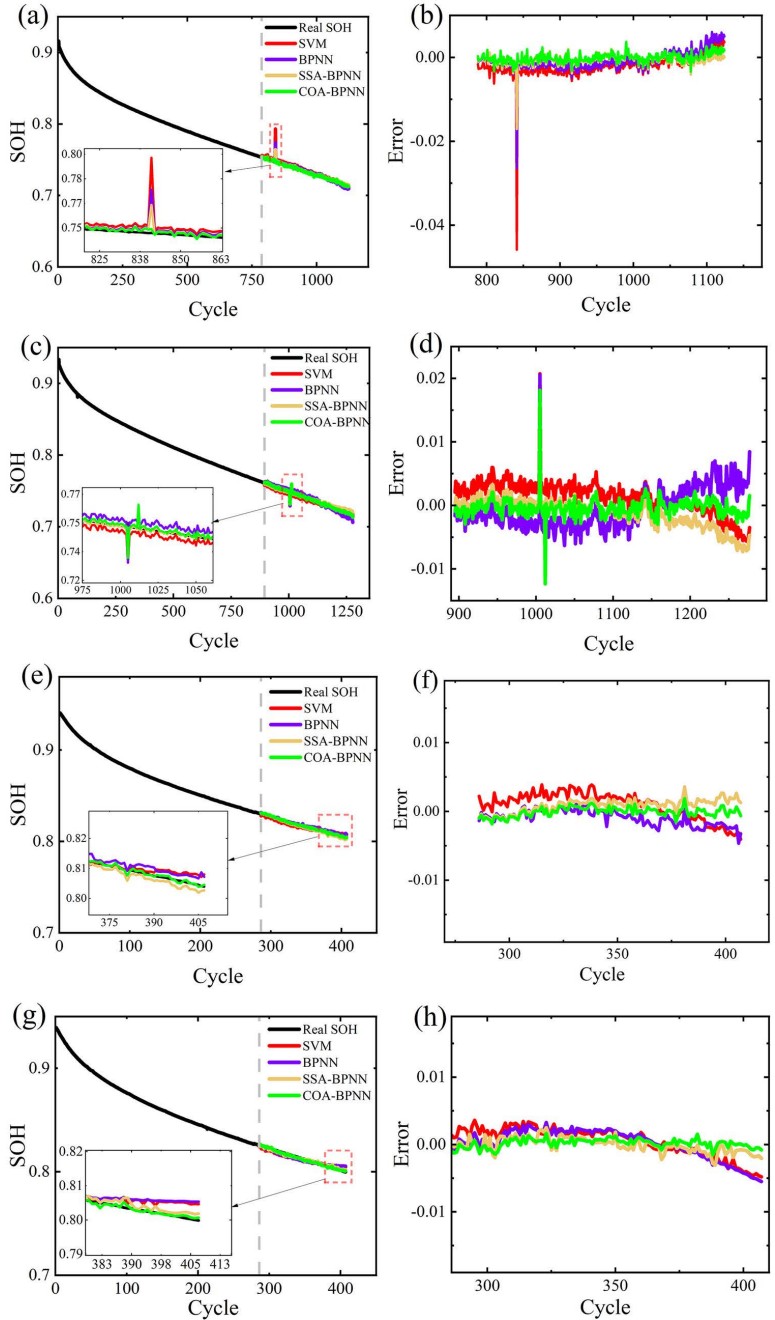

**Fig 4. SOH estimation results and errors.** (a) and (b) represent the estimation results and errors of CY35-0.5/1#1 (c) and (d) represent the estimation results and errors of CY35-0.5/1#2(e) and (f) represent the estimation results and errors of CY45-0.5/1#9, (g) and (h) represent the estimation results and errors of CY45-0.5/1#10.

As shown in Fig 5, for batteries charged at different rates, all four models still achieve good SOH estimation accuracy. Among them, COA-BPNN exhibits the best fitting performance with the smallest errors. For the estimation of Battery CY25-0.25/1#1, COA-BPNN exhibits significant error during the 395th cycle period, similar to the situation observed for Battery CY35-0.5/1#2.

**Table 2. RMSE, MAE and R² of SOH estimation of different models on NCM datasets.**

| Indicators | Models | NCM | | | |
|---|---|---|---|---|---|
| | | #1 | #2 | #9 | #10 |
| RMSE(%) | SVM | 0.37 | 0.29 | 0.21 | 0.22 |
| | BPNN | 0.26 | 0.32 | 0.15 | 0.22 |
| | SSA-BPNN | 0.15 | 0.27 | 0.13 | 0.11 |
| | COA-BPNN | 0.11 | 0.16 | 0.07 | 0.07 |
| MAE(%) | SVM | 0.24 | 0.23 | 0.19 | 0.19 |
| | BPNN | 0.19 | 0.27 | 0.12 | 0.18 |
| | SSA-BPNN | 0.10 | 0.19 | 0.11 | 0.09 |
| | COA-BPNN | 0.09 | 0.10 | 0.05 | 0.05 |
| R² | SVM | 0.9002 | 0.9514 | 0.9192 | 0.9113 |
| | BPNN | 0.9479 | 0.9413 | 0.9632 | 0.9139 |
| | SSA-BPNN | 0.9826 | 0.9581 | 0.9719 | 0.9786 |
| | COA-BPNN | 0.9904 | 0.9851 | 0.9919 | 0.9920 |

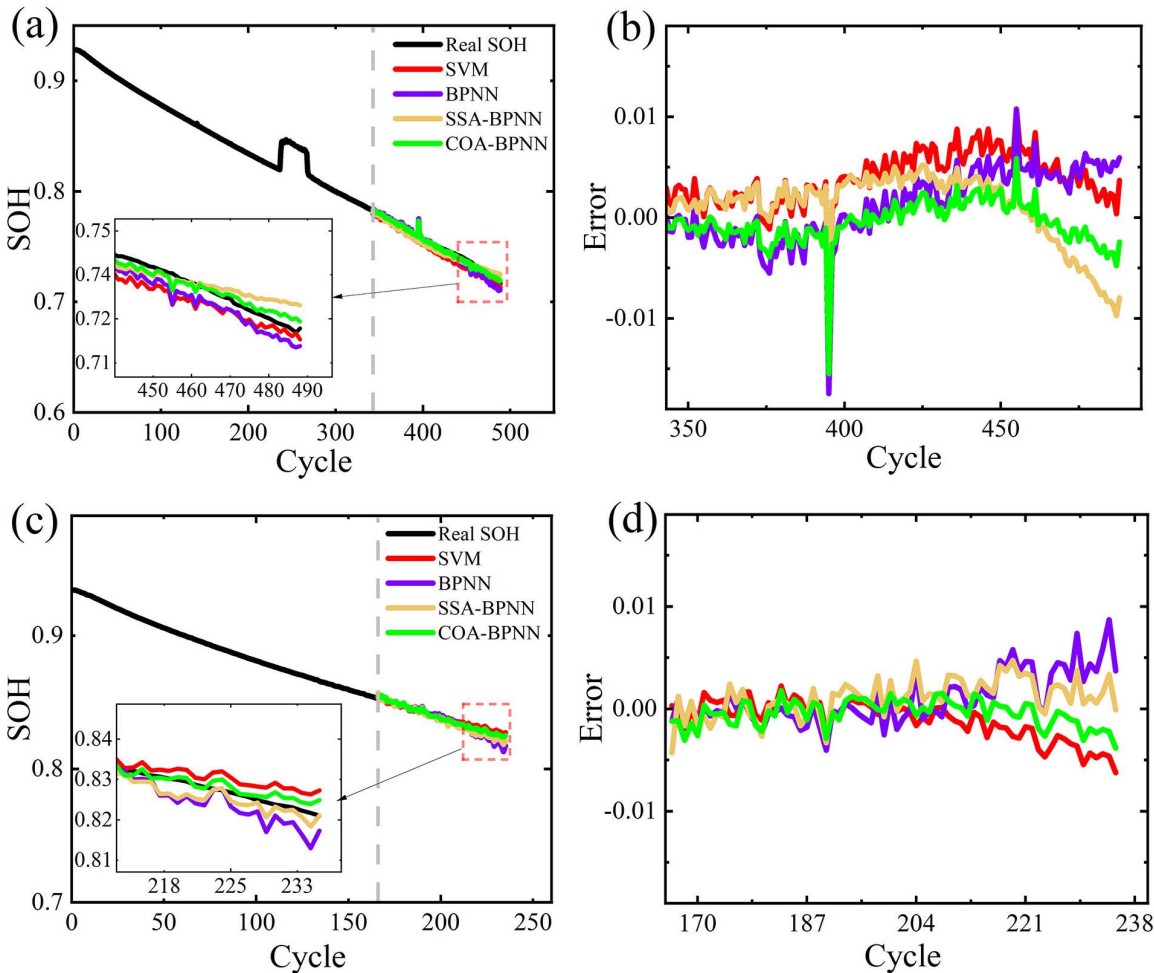

**Fig 5. SOH estimation results and errors.** (a) and (b) represent the estimation results and errors of CY25-0.25/1#1(c) and (d) represent the estimation results and errors of CY25-0.25/1#2.

**Table 3. RMSE, MAE and $R^2$ of SOH estimation of different models on NCA datasets.**

| Models | NCA(#1) | | | NCA(#2) | | |
|---|---|---|---|---|---|---|
| | RMSE(%) | MAE(%) | $R^2$ | RMSE(%) | MAE(%) | $R^2$ |
| SVM | 0.47 | 0.41 | 0.9339 | 0.22 | 0.16 | 0.9433 |
| BPNN | 0.38 | 0.30 | 0.9575 | 0.27 | 0.19 | 0.9136 |
| SSA-BPNN | 0.37 | 0.31 | 0.9594 | 0.21 | 0.17 | 0.9469 |
| COA-BPNN | 0.22 | 0.16 | 0.9852 | 0.14 | 0.11 | 0.9777 |

The comparison results of these batteries in different models are shown in Table 3. As shown in Table 3, the RMSE for four models is less than 0.47%, the MAE is less than 0.41%, and the $R^2$ value is above 0.91. Unlike before, when estimating CY25-0.25/1#2, SVM's estimation results are better than BPNN. The maximum RMSE for BPNN is 0.27%, which is 0.05% higher than SVM. This could be due to the influence of the random initialization of weights and thresholds in BPNN, resulting in network instability. The estimation results of SSA-BPNN are similar to those of COA-BPNN, but the estimation accuracy of the COA-BPNN model is higher. Its RMSE reaches a minimum of 0.14%, MAE reaches a minimum of 0.11%, and $R^2$ reaches a maximum of 0.9852. This indicates the superiority of the COA method.

## Conclusion

The paper proposes a method for estimating the SOH of lithium-ion batteries during the late stage of the CV charging phase. This method is not affected by the randomness of the initial charging stage, thus ensuring stability and accuracy in extracting HF. To better reflect the aging condition of the battery, the current curve during the CV charging phase is converted into a dQ/dI curve, and two health characteristics are extracted from the data within the range of 200mA current variation. Furthermore, to enhance prediction accuracy, COA is utilized to optimize the weights and thresholds of BPNN, thereby addressing the issue of BPNN's initialization parameters. The proposed method has been validated using two publicly available datasets, and the results demonstrate high accuracy in estimating the SOH of batteries at different environmental temperatures and charging rates. This validates the feasibility of estimating the SOH of lithium-ion batteries during the late stage of CV charging and the advantages of COA-BPNN. The estimation results of COA-BPNN are superior to those of SVM, BPNN, and SSA-BPNN, with RMSE and MAE both less than 0.22% and 0.16%, respectively. Although the COA-BPNN method demonstrated promising performance in experimental datasets, its generalization capability still requires further validation across more diverse datasets and application scenarios. Moreover, the SOH of batteries is a complex multidimensional issue, and there may be other important features related to battery aging that have not been extracted. In the future, we will further research and develop more effective feature extraction techniques to uncover more key information related to health status in battery data.

## Supporting information

**S1 File. CY45-0.5/1#9 Code file for data processing and prediction results.**
(RAR)

## Author contributions

**Data curation:** Mingyang Li.

**Formal analysis:** Yanhua Xian.

**Methodology:** Yanhua Xian, Mingyang Li.

**Supervision:** Jiayin Huang.

**Validation:** Jiayin Huang.

**Writing – original draft:** Yanhua Xian, Mingyang Li.

**Writing – review & editing:** Mingyang Li.

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
