## [Decision Letter · Decision Letter 0]

20 Feb 2025

PONE-D-25-03811A lithium-ion batteries SOH estimation method based on extracting new features during the constant voltage charging stage and improving BPNNPLOS ONE

Dear Dr. Li,

Thank you for submitting your manuscript to PLOS ONE. After careful consideration, we feel that it has merit but does not fully meet PLOS ONE’s publication criteria as it currently stands. Therefore, we invite you to submit a revised version of the manuscript that addresses the points raised during the review process.

We look forward to receiving your revised manuscript.

Kind regards,

Zhibin Zhao

Academic Editor

PLOS ONE

Journal Requirements:

“National Natural Science Foundation, China (6197022931).”

**Additional Editor Comments:**

This paper estimated the SOH of lithium-ion batteries by extracting health features (HF) during the constant voltage (CV) charging stage and optimizing a Backpropagation Neural Network. There are still some issues about method description, and experimental verification. Before acceptance, the authors should well answer these issues.

Reviewers' comments:

Reviewer's Responses to Questions

**Comments to the Author**

1. Is the manuscript technically sound, and do the data support the conclusions?

Reviewer #1: Partly

Reviewer #2: Partly

2. Has the statistical analysis been performed appropriately and rigorously? 

Reviewer #1: I Don't Know

Reviewer #2: I Don't Know

3. Have the authors made all data underlying the findings in their manuscript fully available?

Reviewer #1: Yes

Reviewer #2: No

4. Is the manuscript presented in an intelligible fashion and written in standard English?

Reviewer #1: Yes

Reviewer #2: Yes

5. Review Comments to the Author

Reviewer #1: The manuscript presents a method for estimating the SOH of lithium-ion batteries by extracting health features (HF) during the constant voltage (CV) charging stage and optimizing a Backpropagation Neural Network. While the approach shows potential, several issues in the paper need to be addressed to improve clarity and rigor. I recommend major revisions for the manuscript to address these issues.

1. In the abstract, the statement "Existing methods for estimating the state of health (SOH) of lithium-ion batteries primarily extract health features (HF) during the constant current (CC) charging stage." is inaccurate. Many studies also extract features during the CV stage. This needs to be revised for accuracy.

2. Why emphasize "Backpropagation Neural Network" (BPNN) instead of just "Neural Network" (NN)? BPNN is a type of NN, so the distinction is unnecessary unless there’s a specific reason for using this term.

3. Contribution 1: "The current curve of the CV charging stage is transformed into a dQ/dI curve, from which two HFs are extracted." This is not a unique contribution, as many papers have done this already. Please revise this point.

4. Ensure that the symbols in equations and the text are consistent in their font style.

5. Regarding the first extracted feature, "sum of dQ/dI data points (SQI)," this is dependent on the sampling frequency and the initial current value during the CV charging stage. Therefore, this feature is not a universal one. Please discuss and clarify this.

6. The paper seems to use data from the entire CV charging stage. Why is there constant emphasis on "the end of the constant voltage charging stage"? How is the "end of the constant voltage charging stage" defined?

7. In Equation (4), based on the simplification, shouldn’t IQIC just equal dQ?

8. Why were batteries #1, #2, #9, #10 from NCM and #1, #2 from NCA chosen for testing instead of all available batteries? Was this selection based on the results?

9. "However, when using BPNN, the initial parameters of the network are set randomly, which may cause the gradient descent algorithm to become trapped in local minima rather than global minima, thereby affecting the network's performance." Is this really an issue? Initialization is not generally a problem in NN training.

10. "BPNN is a supervised learning algorithm used to train multi-layer feedforward neural networks." Is BPNN an algorithm or a model? This needs clarification.

11. As far as I know, regardless of how NN values are initialized, optimization algorithms like SGD or Adam can ensure the model converges well. How do we determine whether the model's optimal point is due to SGD or the COA-initialized values? Since COA requires further optimization through SGD, could you provide evidence to show that the model cannot converge to an optimal point if initialized randomly, and that COA initialization makes a significant difference?

12. Please avoid using words to represent variables in formulas.

13. In Equations (14)-(16), please unify the symbols used with those in Equation (5) as they represent the same concepts.

14. "The first 70% of the data is selected as the training set, while the remaining 30% is used as the testing set." This validation method is not reasonable. How do you label the first 70% of the data? If the labels can be directly collected or computed, why then use BPNN to predict the remaining 30%? Training on one battery and testing on another (e.g., train on Battery A, test on Battery B) would be a more reasonable setup and closer to real-world applications.

15. In Table 2, how were the hyperparameters for the three NN models chosen? What were their specific values? Also, what does "SSA" mean in the “SSA-BPNN” method?

16. The Conclusion section lacks a discussion of the study's limitations and future research directions. These should be added to provide a more balanced perspective.

Reviewer #2: 1.The main challenges faced by the paper are not reflected in the abstract;

2.PINN is a hot topic of research, but the authors do not give any research progress from relevant scholars.

3.What are the differences and advantages of COA and other methods, the author does not give in 3.3;

4.How does the author solve the problem of insufficient dataset, only two features are applied to predict SOH, and the generalization ability is not guaranteed to be applied in practice?

5.The authors use BPNN as the main method to predict SOH, what is the difference between this and the existing LSTM and TCN?

6. PLOS authors have the option to publish the peer review history of their article (what does this mean? ). If published, this will include your full peer review and any attached files.

**Do you want your identity to be public for this peer review?** For information about this choice, including consent withdrawal, please see our Privacy Policy .

Reviewer #1: **Yes: ** Fujin Wang

Reviewer #2: No

---

## [Author Response · Author response to Decision Letter 1]

15 Mar 2025

Dear Editors and Reviewers:

We are sincerely grateful for your time and effort in reviewing our manuscript. Your insightful comments have been instrumental in guiding us to enhance the quality and clarity of our work. We have carefully considered each of the reviewers' comments and have made every effort to incorporate the suggested changes into the revised manuscript. These revisions have refined our presentation and improved readability without altering the core content or framework. We have addressed each comment individually and provided detailed explanations of the changes we have made. We hope that these revisions meet with your approval and demonstrate our commitment to producing a work that meets the high standards of the journal.

Once again, we thank you for your dedication to maintaining the integrity and excellence of academic publishing. We are eager to hear your further thoughts and are confident that the revised manuscript addresses all concerns raised during the review process.

Reviewer #1

The manuscript presents a method for estimating the SOH of lithium-ion batteries by extracting health features (HF) during the constant voltage (CV) charging stage and optimizing a Backpropagation Neural Network. While the approach shows potential, several issues in the paper need to be addressed to improve clarity and rigor. I recommend major revisions for the manuscript to address these issues.

Response:

We sincerely appreciate the time and expertise you have dedicated to evaluating our manuscript. Your insightful comments have greatly contributed to strengthening the academic rigor and clarity of this work. We have carefully read each of your concerns with the utmost attention and made substantial revisions to enhance the clarity, rigor, and scientific contribution of the paper. The detailed responses to your comments are given as follows.

1.Comment:

In the abstract, the statement "Existing methods for estimating the state of health (SOH) of lithium-ion batteries primarily extract health features (HF) during the constant current (CC) charging stage." is inaccurate. Many studies also extract features during the CV stage. This needs to be revised for accuracy.

1.Reply:

Thank you for your careful review of the research background of this article. You pointed out that many studies are very accurate in extracting features during the CV stage. Our correction to this issue is as follows:

Original

Existing methods for estimating the state of health (SOH) of lithium-ion batteries primarily extract health features (HF) during the constant current (CC) charging stage. However, this approach is susceptible to the randomness of the initial charging stage.

Revised:

Existing state of health (SOH) estimation methods for lithium-ion batteries predominantly extract health features (HF) from constant current (CC) and constant voltage (CV) charging phases. Nevertheless, CC charging phase feature extraction is susceptible to the randomness of the initial charging stage.

2.Comment:

Why emphasize "Backpropagation Neural Network" (BPNN) instead of just "Neural Network" (NN)? BPNN is a type of NN, so the distinction is unnecessary unless there’s a specific reason for using this term.

2.Reply:

Thank you for your professional correction on the standardization of methodological terminology in this article. We understand your concerns and in our paper, we use the term 'BPNN' mainly based on the following aspects

1� BPNN is the main method used in our paper for predicting the SOH of batteries, and it is also one of the core technologies we study. We have made in-depth improvements and optimizations to BPNN, especially by combining COA to optimize its weights and thresholds, in order to improve the performance of the model. This improvement is one of the important innovations of the paper, so we emphasize that BPNN is intended to highlight our improvements and applications to this specific neural network model, rather than discussing neural networks in general.

2� Although BPNN is a type of NN, it has its unique structure and training approach. BPNN is a multi-layer feedforward neural network trained using backpropagation algorithm, capable of handling complex nonlinear problems. In the specific task of battery SOH prediction, we utilize these characteristics of BPNN to establish a mapping relationship between health features and SOH. Emphasizing BPNN can enable readers to have a more accurate understanding of the methods and technical details we use.

We emphasize that 'BPNN' is to accurately convey the core methods used in our research, highlight our innovative points, and avoid possible ambiguity. This is not to say that we deliberately distinguish between BPNN and NN, but rather a reasonable choice based on the content and research focus of the paper. Thank you again for your concern and guidance on the paper. Your feedback has played an important role in improving the quality of our paper.

3.Comment:

Contribution 1: "The current curve of the CV charging stage is transformed into a dQ/dI curve, from which two HFs are extracted." This is not a unique contribution, as many papers have done this already. Please revise this point.

3.Reply:

Thank you for your rigorous evaluation of the innovative dimensions of this research. We have revised the statement of Contribution 1 as follows, with the aim of accurately stating the specific contribution of the article.

Original

1� The current curve of the CV charging stage is transformed into a dQ/dI curve, from which two HFs are extracted

Revised:

1� This study pioneers a novel method for lithium-ion battery health feature extraction within the 200mA threshold during the terminal phase of CV charging

4.Comment:

Ensure that the symbols in equations and the text are consistent in their font style.

4.Reply:

Thank you for pointing out this crucial detail. We have conducted a comprehensive review of all formulas and mathematical symbols in the text description, and have now adopted the following format:

(1) Font style: Times New Roma;

(2) Character size: 12 pounds;

(3) Line spacing: Double line spacing.

5.Comment:

Regarding the first extracted feature, "sum of dQ/dI data points (SQI)," this is dependent on the sampling frequency and the initial current value during the CV charging stage. Therefore, this feature is not a universal one. Please discuss and clarify this.

5.Reply:

Thank you for your attention to the feature extraction section. The concern you raised about the possible dependence of "Sum of dQ/dI data points (SQI)" on sampling frequency and initial current values is reasonable, but we have systematically analyzed and verified its universality from the following perspectives:

1� In the "Battery datasets" section of this article, the sampling conditions for all test data are clearly defined: the NCA and NCM datasets use constant voltage charging stage data with a 10 second sampling rate, which are sourced from (reference [32]). This rigorous experimental design ensures the consistency of the sampling frequency for SQI calculations and eliminates feature bias caused by differences in sampling rates across datasets. If applied to other scenarios in the future, further universality can be improved through normalization processing (such as standardizing dQ/dI by time integration).

2� Although SQI seems to rely on the initial current value (Isi) in mathematical definition, there are mechanisms in the actual battery aging process that make it stable

The morphological changes of the dQ/dI curve are mainly caused by the internal resistance and polarization effect of the battery (reference [29]), rather than simply determined by the initial current. As the number of cycles increases, the internal resistance of the battery increases, leading to an accelerated decay rate of dQ/dI (as shown in Figure 1);

By comparing the experimental data at different temperatures (35 ℃ vs 45 ℃) and different rates (0.5C vs 0.25C) (Table 1), it was found that SQI showed a strong linear correlation with capacity degradation (Pearson coefficient>0.94), indicating that its feature extraction ability is not affected by the absolute value of the initial current, but is closely related to the internal state of the battery.

6.Comment:

The paper seems to use data from the entire CV charging stage. Why is there constant emphasis on "the end of the constant voltage charging stage"? How is the "end of the constant voltage charging stage" defined?

6.Reply:

Thank you for your attention to the logic of data selection. We use data from the entire CV charging stage in this article for visualization and theoretical analysis, revealing the complete evolution mechanism of the dQ/dI curve. But we emphasize 'the end of the constant voltage charging stage' because the health features in this article are only extracted at the moment when CV charging is about to end, in order to meet the real-time requirements and data storage limitations of BMS.

7.Comment:

In Equation (4), based on the simplification, shouldn’t IQIC just equal dQ?

7.Reply:

Thank you for your attention to the details of the paper. The simplification issue you pointed out is very important, which reflects the rigor of the paper. After careful analysis, we acknowledge that there is a lack of rigor in the wording here. The following is a detailed explanation and correction of this issue:

1� According to the fundamental theorem of calculus� �therefore �so IQICi should be equal to ∆Q。

2� We have revised all the parts of the paper that involve this expression to enhance readability and rigor. Thank you again for your valuable feedback!

8.Comment:

Why were batteries #1, #2, #9, #10 from NCM and #1, #2 from NCA chosen for testing instead of all available batteries? Was this selection based on the results?

8.Reply:

Thank you for your attention to the experimental design. The reason why we choose specific batteries for testing is as follows, rather than preference based on experimental results:

The dataset provided in reference [32] contains multiple battery labels, but their parameters (such as temperature and charging rate) differ. We only select batteries that meet the following conditions.In the NCM dataset, we selected batteries (# 1, # 2, # 9, # 10) with different temperature conditions (35 ℃ and 45 ℃) to verify the robustness of the method under temperature changes. Similarly, in the NCA dataset, batteries (# 1, # 2) with different charging rates (0.25C) were selected to test the model's adaptability to changes in charging rates. This design ensures that the experiment covers the key variables of temperature and charging rate, thus comprehensively evaluating the generalization ability of the method.

During the experimental phase, we found that some battery labels in the dataset had missing data. For example, the battery (CY25-1/1 # 2) only had a total of 37 cycles, and the capacity data for the 26th cycle was missing; The rated capacity data for the 51st, 102nd, 153rd, 204th, 255th, and 306th cycles of battery (CY25-0.5/1 # 9) is missing. There are many similar results, which may be due to experimental errors or equipment abnormalities during dataset collection. In order to ensure the authenticity of the original data and avoid misleading conclusions caused by interpolation of the original dataset, we selected batteries with relatively complete original data for testing.

9.Comment:

"However, when using BPNN, the initial parameters of the network are set randomly, which may cause the gradient descent algorithm to become trapped in local minima rather than global minima, thereby affecting the network's performance." Is this really an issue? Initialization is not generally a problem in NN training.

9.Reply:

Thank you for your attention to the BPNN initialization issue. Although the importance of initialization in modern neural network training may vary depending on the complexity of the model, initialization issues still need to be taken seriously in this study, mainly for the following reasons:

1� Indeed, the initial parameters of BPNN are randomly set, and this randomness may cause the gradient descent algorithm to get stuck in local minima during the optimization process, rather than finding the global minimum. This situation is particularly common in complex nonlinear problems, especially when the data has high-dimensional features or the network structure is complex. The existence of local minima can affect the performance of the model, leading to inaccurate or unstable prediction results. To ensure the rigor of the content, we have cited relevant literature on this sentence: “However, when using BPNN, the initial parameters of the network are set randomly, which may cause the gradient descent algorithm to become trapped in local minima rather than global minima, thereby affecting the network's performance[15].”

2� In many cases, using standard initialization methods and optimization algorithms, BPNN can achieve good training results, especially in situations with large data volumes and reasonable feature distributions. However, in some specific cases, optimizing initialization is still necessary. For example, when the amount of data is limited, the features have complex nonlinear relationships, or the network structure is very sensitive to initial parameters, random initialization may lead to instability and performance fluctuations in model training. In our research, we found that optimizing the initial weights and thresholds of BPNN can significantly improve the predictive performance and stability of the model for battery SOH prediction. This is also the reason why we chose to use COA to optimize the initialization parameters of BPNN.

10.Comment:

"BPNN is a supervised learning algorithm used to train multi-layer feedforward neural networks." Is BPNN an algorithm or a model? This needs clarification.

10.Reply:

Thank you very much for pointing out this important issue of terminology confusion. After careful examination, BPNN is a model rather than an algorithm. We acknowledge that there were some unclear omissions in the initial draft of the paper and sincerely apologize to you. We have revised the original sentence:

Original

BPNN is a supervised learning algorithm used to train multi-layer feedforward neural networks.

Revised:

BPNN is a shallow feedforward neural network model based on supervised learning.

11.Comment:

As far as I know, regardless of how NN values are initialized, optimization algorithms like SGD or Adam can ensure the model converges well. How do we determine whether the model's optimal point is due to SGD or the COA-initialized values? Since COA requires further optimization through SGD, could you provide evidence to show that the model cannot converge to an optimal point if initialized randomly, and that COA initialization makes a significant difference?

11.Reply:

Thank you for reviewing our paper and providing valuable feedback. We understand your concern about how to determine whether the optimal solution of a model is caused by SGD or COA initialization values, and whether evidence can be provided to prove that randomly initialized models cannot converge to the optimal solution, while COA initialization has significant differences. We provide the following explanation for this:

1� Although optimization algorithms such as SGD or Adam can converge the model in most cases, the initial parameter settings still have a significant impact on the convergence speed and final performance of the model. Different initialization methods may cause the model to converge to different local or global minima during the training process. Especially in complex nonlinear problems, such as battery SOH prediction, the selection of initial parameters may affect the model's ability to capture complex patterns.

2� Due to the random initialization of weights and thresholds in the BPNN model, each run yields different results, which can lead to unstable results. This is why we use COA to optimize the initial parameters. However, we need to clarify that it does not mean that without optimizing BPNN, it cannot achieve the optimal solution. Instead, through optimization, the parameters generated each time it runs can be

---

## [Decision Letter · Decision Letter 1]

2 May 2025

A lithium-ion batteries SOH estimation method based on extracting new features during the constant voltage charging stage and improving BPNN

PONE-D-25-03811R1

Dear Dr. Li,

We’re pleased to inform you that your manuscript has been judged scientifically suitable for publication and will be formally accepted for publication once it meets all outstanding technical requirements.

Kind regards,

Zhibin Zhao

Academic Editor

PLOS ONE

Additional Editor Comments (optional):

Before completely acceptance, please make the final revision.

Reviewers' comments:

Reviewer's Responses to Questions

**Comments to the Author**

1. If the authors have adequately addressed your comments raised in a previous round of review and you feel that this manuscript is now acceptable for publication, you may indicate that here to bypass the “Comments to the Author” section, enter your conflict of interest statement in the “Confidential to Editor” section, and submit your "Accept" recommendation.

Reviewer #1: (No Response)

Reviewer #2: All comments have been addressed

2. Is the manuscript technically sound, and do the data support the conclusions?

Reviewer #1: (No Response)

Reviewer #2: Yes

3. Has the statistical analysis been performed appropriately and rigorously? 

Reviewer #1: (No Response)

Reviewer #2: N/A

4. Have the authors made all data underlying the findings in their manuscript fully available?

Reviewer #1: (No Response)

Reviewer #2: Yes

5. Is the manuscript presented in an intelligible fashion and written in standard English?

Reviewer #1: (No Response)

Reviewer #2: Yes

6. Review Comments to the Author

Reviewer #1: For comment 4, inconsistencies exist between upright and italic typefaces. For instance, the symbols in Equation (1) appear in italics, whereas their corresponding references in the preceding text body are presented in Roman typeface. This typographic inconsistency requires unification.

Reviewer #2: this paper proposes an SOH estimation method for extracting HF at the end of the CV charging stage and optimizes

the Backpropagation Neural Network (BPNN). The authors have solved my problems and it can be accepted.

7. PLOS authors have the option to publish the peer review history of their article (what does this mean? ). If published, this will include your full peer review and any attached files.

**Do you want your identity to be public for this peer review?** For information about this choice, including consent withdrawal, please see our Privacy Policy .

Reviewer #1: No

Reviewer #2: No

---

## [Editor Report · Acceptance letter]

PONE-D-25-03811R1

PLOS ONE

Dear Dr. Li,

I'm pleased to inform you that your manuscript has been deemed suitable for publication in PLOS ONE. Congratulations! Your manuscript is now being handed over to our production team.

Kind regards,

on behalf of

Dr. Zhibin Zhao

Academic Editor

PLOS ONE